



**Inter-comparison and improvement of 2-stream shortwave radiative transfer**
**models for unified treatment of cryospheric surfaces in ESMs**
Cheng Dang[1], Charles S. Zender[1], Mark G. Flanner[2]
[1]Department of Earth System Science, University of California, Irvine, CA, USA
[2] Department of Climate and Space Sciences and Engineering, University of Michigan,
Ann Arbor, MI, USA :
*Correspondence to*: Cheng Dang (cdang5@uci.edu)
**Abstract.** Snow is an important climate regulator because it greatly increases the surface
albedo of large parts of the Earth. Earth System Models (ESMs) often adopt 2-stream
approximations with different radiative transfer techniques, the same snow therefore has
different solar radiative properties depending whether it is on land or on sea ice. Here we
inter-compare three 2-stream algorithms widely used in snow models, improve their
predictions at large zenith angles, and introduce a hybrid model suitable for all
cryospheric surfaces in ESMs. The algorithms are those employed by the SNow ICe and
Aerosol Radiative (SNICAR) module used in land models, and by Icepack, the column
physics used in the Los Alamos sea ice model CICE and MPAS-seaice, and a 2-stream
discrete ordinate (2SD) model. Compared with a 16-stream benchmark model, the errors
in snow visible albedo for a direct-incident beam from all three 2-stream models are
small (<±0.005) and increase as snow shallows, especially for aged snow. The errors in
direct near-infrared (near-IR) albedo are small (<±0.005) for solar zenith angles $\theta$ < 75°,
and increase as $\theta$ increases. For diffuse incidence under cloudy skies, Icepack produces
the most accurate snow albedo for both visible and near-IR (<±0.0002) with the lowest
underestimate (-0.01) for melting thin snow. SNICAR performs similarly to Icepack for
visible albedos, with a slightly larger underestimate (-0.02), while it overestimates the
near-IR albedo by an order of magnitude more (up to 0.04). 2SD overestimates both
visible and near-IR albedo by up to 0.03. We develop a new parameterization that adjusts
the underestimated direct near-IR albedo and overestimated direct near-IR heating
persistent across all 2-stream models for solar zenith angles > 75°. These results are
incorporated in a hybrid model SNICAR-AD, which can now serve as a unified solar
radiative transfer model for snow in ESM land, land ice, and sea-ice components.





## 1. Introduction

Snow cover on land, land ice, and sea ice, modulates the surface energy balance of large parts of the Earth, principally because even a thin layer of snow greatly increases the surface albedo. Integrated over the solar spectrum, the broadband albedo of opaque snow ranges from 0.7 – 0.9 (e.g., Wiscombe and Warren 1980; Dang et al., 2015). In contrast, the albedo of other natural surfaces is smaller: 0.2, 0.25, and 0.5-0.7 for damp soil, grassland, and bare multi-year sea ice, respectively (Perovich 1996; Liang et al., 2002; Brandt et al., 2005; Bøggild et al., 2010). An accurate simulation of the shortwave radiative properties of snowpack is therefore crucial for spectrally partitioning solar energy and representing snow-albedo feedbacks across the Earth system. Unfortunately, computational demands and coupling architectures often constrain representation of snowpack radiative processes in Earth System Models (ESMs) to relatively crude approximations such as 2-stream methods (Wiscombe and Warren, 1980, Toon et al., 1989). In this work, we inter-compare 2-stream methods widely used in snow models and then introduce a new parameterization that significantly reduces their snowpack reflectance and heating biases at large zenith angles, to produce more realistic behavior in polar regions.

Snow albedo is determined by many factors including the snow grain radius, the solar zenith angle, cloud transmittance, light-absorbing particles, and the albedo of underlying ground if snow is optically thin (Wiscombe and Warren, 1980; Warren and Wiscombe, 1980); it also varies strongly with wavelength since the ice absorption coefficient varies by 7 orders of magnitudes across the solar spectrum (Warren and Brandt, 2008). At visible wavelengths (0.2 - 0.7 μm), ice is almost non-absorptive so that the absorption of visible energy by snowpack is mostly due to the light-absorbing particles (e.g. black carbon, organic carbon, mineral dust) that were incorporated during ice nucleation in clouds, scavenged during precipitation, or slowly sedimented from the atmosphere by gravity (Warren and Wiscombe, 1980, 1985; Doherty et al., 2010, 2014, 2016; Wang et al., 2013; Dang and Hegg 2014). As snow becomes shallower, visible photons are more likely to penetrate through snowpack and get absorbed by darker underlying ground. At near-infrared (near-IR) wavelengths (0.7 – 5 μm), ice is much more absorptive and the snow albedo is lower than the visible albedo. Larger ice crystals form a lower albedo surface than smaller ice crystals hence aged snowpacks absorb more solar energy. Photons incident at smaller solar zenith angles are more likely to penetrate deeper vertically and be scattered in the snowpack until being absorbed by the ice/the





underlying ground/absorbing impurities, which also leads to a smaller snow albedo.
To compute the reflected solar flux, spectrally resolved albedo must be weighted by the
incident solar flux, which is mostly determined by solar zenith angle, cloud cover and
transmittance, and column water vapor. Modeling the solar properties of snowpacks must
consider the spectral signatures of these atmospheric properties.
Several parameterizations have been developed to compute the snow solar properties
without solving the radiative transfer equations and some are incorporated into ESMs or
regional models. Marshall and Warren (1987) and Marshall (1989) parameterized snow
albedo in both visible and near-IR bands as functions of snow grain size, solar zenith
angle, cloud transmittance, snow depth, underlying surface albedo, and black carbon
content. Marshall and Oglesby (1994) used this in an ESM. Gardner and Sharp (2010)
parameterized the all-wave snow albedo with similar inputs. This was incorporated into
the regional climate model RACMO
(https://www.projects.science.uu.nl/iceclimate/models/racmo.php)    to    simulate    snow
albedo in glaciered regions like Antarctica and Greenland (Munneke et al., 2011). Dang
et al., (2015) compute snow albedo as functions of snow grain radius, black carbon
content, and dust content for visible and near-IR bands and 14 narrower bands used in the
rapid    radiative    transfer    model    (RRTM,    Mlawer    and    Clough,    1997).    Their
parameterization can also be expanded to different solar zenith angles using the zenith
angle parameterization developed by Marshall and Warren (1987). Aoki et al., (2011)
developed a more complex model (PBSAM) based on the offline snow albedo and a
transmittance look-up table. This can be applied to multilayer snowpack to compute the
snow albedo and the solar heating profiles as functions of snow grain size, black carbon
and    dust    content,    snow    temperature,    and    snowmelt    water    equivalent.    These
parameterizations are often in the form of simplified polynomial equations, and are
especially suitable to long-term ESM simulations that require less time-consuming snow
representations.
More complex models that explicitly solve the multiple scattering radiative transfer
equations have also been developed to compute snow solar properties. Flanner and
Zender (2005) developed the SNow Ice and Aerosol Radiation model (SNICAR) that
utilizes 2-stream approximations (Wiscombe and Warren 1980; Toon et al., 1989) to
predict heating and reflectance for multi-layer snowpack. They implemented SNICAR in
the Community Land Model (CLM) to predict snow albedo and vertically-resolved solar
absorption for snow-covered surfaces. Before SNICAR, CLM prescribed snow albedo





and confined all solar absorption to the top snow layer (Flanner and Zender 2005). Over
the past decades, updates and new features have been added to SNICAR to consider more
processes such as black carbon/ice mixing states (Flanner et al., 2012) and snow grain
shape (He et al., 2018b). Concurrent with the development of SNICAR, Briegleb and
Light (2007) improved the treatment of sea-ice solar radiative calculations in Community
Climate System Model (CCSM). They implemented a 2-stream delta-Eddington method
that allows CCSM to compute bare/ponded/snow-covered sea ice albedo and solar
absorption profiles of multi-layer sea ice. Before these improvements, the sea-ice albedo
was computed based on surface temperature, snow thickness, and sea-ice thickness using
averaged sea ice and snow albedo. This method has carried into the sea-ice physics
library Icepack (https://github.com/CICE-Consortium/Icepack/wiki) that comprises the
column physics used by the Los Almos Sea Ice Model CICE (Hunke et al., 2010) and
MPAS-seaice (Turner et al., 2018). CICE itself is used in numerous global and regional
models.

The shortwave methods in SNICAR and in CICE solve the multiple scattering radiative
transfer equations and provide much improved solar radiative representations for the
cryosphere, though their separate development and implementation created an artificial
divide for snow simulation. In ESMs that utilize both SNICAR and CICE/MPAS-seaice,
such as the Community Earth System Model (CESM, http://www.cesm.ucar.edu/) and the
Energy Exscale Earth System Model (E3SM, previously known as ACME,
https://e3sm.org/), the solar radiative properties of snow on land and snow on sea ice are
computed separately via SNICAR and CICE/MPAS-seaice. As a result, the same snow in
nature has different solar radiative properties such as reflectance depending on which
model represents it. These differences are model artifacts that should be eliminated so
that snow has consistent properties across the Earth system.

In this paper, we evaluate the accuracy and biases of three 2-stream algorithms described
in Section 2 and Table 1, including the algorithms used in SNICAR and Icepack, at
representing reflectance and heating. We use these results to develop and justify a unified
surface shortwave radiative transfer method for all Earth system model components in the
cryosphere.

**2. Radiative Transfer Model**

In this section, we summarize the three 2-stream models and the benchmark DISORT





model with 16-streams. These algorithms are well documented in papers by Toon et al.,
(1989), Briegleb and Light (2007), Jin and Stamnes (1994), and Stamnes et al. (1988).
Readers interested in detailed mathematical derivations should refer to those papers. We
only include their key equations to illustrate the difference among 2-stream models for
discussion purposes.

## 2.1 SNICAR
SNICAR adopts the 2-stream algorithms and the rapid solver developed by Toon et al.,
(1989) to compute the solar properties of multi-layer snowpacks. These 2-stream
algorithms are derived from the general equation of radiative transfer in a plane parallel
media:

$\mu \frac{\partial I}{\partial \tau}(\tau, \mu, \Phi) = I(\tau, \mu, \Phi) - \frac{\varpi}{4\pi} \int_0^{2\pi} \int_{-1}^{1} P(\mu, \mu', \phi, \phi') I(\tau, \mu', \Phi') d\mu' d\phi' - S(\tau, \mu, \Phi)$

157    (1)



where $arccos(\mu)$ and $\Phi$ are zenith angle and azimuth angle, $\varpi$ is single-scattering albedo.
On the right-hand side, the three terms are intensity at optical depth $\tau$, internal source
term due to multiple scattering, and external source term $S$. For a purely external source
at solar wavelengths $S$ is:

$S = \frac{\varpi}{4} F_s P(\mu, -\mu_0, \phi, \phi_0) exp\left(\frac{-\tau}{\mu_0}\right)$    (2)

where $\pi F_s$ is incident solar flux, $\mu_0$ is the incident direction of the solar beam. Integrating
equation (1) over azimuth and zenith angles yields the general solution of 2-stream
approximations (Meador and Weaver, 1980). The upward and downward fluxes at optical
depth $\tau$ of layer n can be represented as:


$F_n^+ = k_{1n} exp(\Lambda_n \tau) + \Gamma_n k_{2n} exp(-\Lambda_n \tau) + C_n^+(\tau)$    (3a)

$F_n^- = \Gamma_n k_{1n} exp(\Lambda_n \tau) + k_{2n} exp(-\Lambda_n \tau) + C_n^-(\tau)$    (3b)


where $\Lambda_n, \Gamma_n C_n$ are known coefficients determined by the 2-stream method, incident
solar flux, and solar zenith angle; whereas $k_{1n}$ and $k_{2n}$ are unknown coefficients



determined by the boundary conditions. For an N-layer snowpack, the solutions for
upward and downward fluxes are coupled at layer interfaces to generate 2N equations
with 2N unknown coefficients $k_{1n}$ and $k_{2n}$. Combining these equations linearly generates
a new set of equations with terms in tridiagonal form that enables the application of a fast
tri-diagonal matrix solver. With the solved coefficients, the upward and downward fluxes
are computed at different optical depths (Equations 3a and 3b) and eventually the
reflectance, transmittance, and absorption profiles of solar flux for any multilayer
snowpack.
SNICAR itself implements all three 2-stream algorithms in Toon et al., (1989):
Eddington, Quadrature, and Hemispheric-mean. In ESM simulations, it utilizes the
Eddington and Hemispheric-mean approximations to compute the visible and near-IR
snow properties, respectively (Flanner et al., 2007). In addition to their algorithms,
SNICAR implements a Delta-transform of the fundamental input variables asymmetry
factor ($g$), single-scattering albedo ($\varpi$), and optical depth ($\tau$) to account for the strong
forward scattering in snow (Equations 2 (a)-(c), Wiscombe and Warren, 1980).
2.2. Icepack, CICE, and MPAS-seaice
Icepack, CICE and MPAS-seaice use the same solar radiative treatment developed and
documented by Briegleb and Light (2007). In the following discussions, we will refer to
this method as CICE since it is more widely used. Sea ice is divided into multiple layers
to first compute the single-layer reflectance and transmittance using 2-stream delta-
Eddington solutions to account for the multiple scattering of light within each layer
(Equation set 50, Briegleb and Light, 2007), where the name "delta" implies CICE
implements the Delta-transform to account for the strong forward scattering of snow and
sea ice (Equations 2 (a)-(c), Wiscombe and Warren, 1980). The direct albedo and
transmittance are computed by equations:
$R(\mu_{0,n}) = A_n \, exp\left(\frac{-\tau}{\mu_{0,n}}\right) + B_n(exp(\varepsilon_n \tau) - exp(-\varepsilon_n \tau)) - K_n$                    (4a)
$T(\mu_{0,n}) = E_n + H_n(exp(\varepsilon_n \tau) - exp(-\varepsilon_n \tau)) \, exp\left(\frac{-\tau}{\mu_{0,n}}\right)$                    (4b)
where coefficients $A_n$, $B_n$, $K_n$, $E_n$, $H_n$, and $\varepsilon_n$ are determined by the single-scattering
albedo ($\varpi$), asymmetry factor ($g$), optical depth ($\tau$), and angle of incident beam at layer n
($\mu_{0,n}$). Following the delta-Eddington assumption, simple formulas are available for the
single-layer reflectance and transmittance under both clear sky (direct flux, equations 4a
and 4b) and overcast sky (diffuse flux) conditions, however, the formula derived by
applying diffuse-flux upper boundary conditions sometimes yields negative albedos
(Wiscombe 1977). To avoid the unphysical values, diffuse reflectance $\bar{R}$ and
transmittance $\bar{T}$ of a single layer are computed by integrating the direct reflectance $R(\mu)$
and transmittance $T(\mu)$ over the incident hemisphere assuming isotropic incidence:

$\bar{R} = 2 \int_0^1 \mu R(\mu) d\mu$                                              (5a)

$\bar{T} = 2 \int_0^1 \mu T(\mu) d\mu$                                              (5b)

This is the same as the method proposed by Wiscombe and Warren (1980, their equation
5). In practice, eight Gaussian angles are implemented to perform the integration for
every layer.

These layer reflectance and transmittance of direct and diffuse components are then
combined to account for the inter-layer scattering of light to compute the reflectance and
transmission at every interface (Equation set 51, Briegleb and Light, 2007), and
eventually the upward and downward fluxes (Equation set 52, Briegleb and Light, 2007).
These upward and downward fluxes at each optical depth are then used to compute the
column reflectance and transmittance, and the absorption profiles for any multilayered
media, such as snowpacks on land and sea ice.

In nature, a large fraction of sea ice is covered by snow during winter. As snow melts
away in late spring and summer, it exposes bare ice, and melt ponds form on the ice
surface. Such variation of sea-ice surface types requires the shortwave radiative transfer
model to be flexible and capable of capturing the light refraction and reflection.
Refractive boundaries exist where air (refractive index $m_{re} = 1.0$), snow (assuming snow
as medium of air containing a collection of ice particles, $m_{re} = 1.0$), pond (assuming pure
water, $m_{re} = 1.33$), and ice (assuming pure ice, $m_{re} = 1.31$) are present in the same sea-ice
column. The general solution of delta-Eddington, and the 2-stream algorithms used in
SNICAR are not applicable to such non-uniformly refractive layered media. To include
the effects of refraction, Briegleb and Light (2007) modified the adding formula at the
refractive boundaries (i.e. interfaces between air/ice, snow/ice, air/pond). The reflectance
and transmittance of the adjacent layers above and below the refractive boundary are



combined with modifications to include the Fresnel reflection and refraction of direct and
diffuse fluxes (Section 4.1, Briegleb and Light, 2007). This adding-doubling delta-
Eddington method can thus be applied to any layered media with either uniform (e.g.,
snow on land) or non-uniform (e.g., snow on sea ice) refractive indexes.
In this paper, we focus on snowpacks that can be treated as uniform refractive media such
as the air/snowpack/land columns assumed in SNICAR. An ideal radiative treatment for
snow should however keep the potential to include refraction for further applications to
snow on sea ice or ice sheets. Therefore, besides these two widely used algorithms in
Icepack and SNICAR, we evaluate a third algorithm (section 2.3) that can be applied to
layered media with either uniform or non-uniform refractive indexes.
2.3. 2-stream discrete-ordinate algorithm (2SD)
A refractive boundary also exists between the atmosphere and the ocean, and models
have been developed to solve the radiative transfer problems in the atmosphere-ocean
system using the discrete-ordinate technique (e.g. Jin and Stamnes, 1994; Lee and Liou,
2007). Similar to the 2-stream algorithms of Toon et al., (1989) used in SNICAR, Jin and
Stamnes (1994) also developed their algorithm from the general equation:

$$\mu \frac{\partial I}{\partial \tau}(\tau, \mu) = I(\tau, \mu) - \frac{\varpi}{4\pi} \int_{-1}^{1} P(\tau, \mu, \mu') I(\tau, \mu') d\mu' - S(\tau, \mu) \tag{6}$$

Equation (6) is the azimuthally integrated version of equation (1). However, for vertically
inhomogeneous media like the atmosphere-ocean or sea ice, the external source term
$S(\tau, \mu)$ is different. Specifically, for the medium of total optical depth $\tau^a$ above the
refractive interface, one must consider the contribution from the upward beam reflected
at the refractive boundary (second term on the right-hand side):

$$S^a(\tau, \mu) = \frac{\varpi}{4\pi} F_s P(\tau, -\mu_0, \mu) \, exp\left(\frac{-\tau}{\mu_0}\right) + \frac{\varpi}{4\pi} F_s R(-\mu_0, m) P(\tau, +\mu_0, \mu) \, exp\left(\frac{-(2\tau^a - \tau)}{\mu_0}\right)$$
279                                                                                            (7)

where $R(-\mu_0, m)$ is the Fresnel reflectance of radiation and $m$ is the ratio of the
refractive indices of the lower to the upper medium. For the medium below the refractive
interface, one must account for the Fresnel transmittance $T(-\mu_0, m)$ and modify the
angle of beam travel in media b:






$$S^b(\tau,\mu) = \frac{\varpi}{4\pi} \frac{\mu_0}{\mu_{0n}} F_s T(-\mu_0,m) P(\tau,-\mu_0,\mu) \, exp\left(\frac{-\tau^a}{\mu_0}\right) exp\left(\frac{-(\tau-\tau^a)}{\mu_{0n}}\right) \qquad (8)$$

where $\mu_{0n}$ is the cosine zenith angle of refracted beam incident at angle $\mu_0$ above
refractive boundary, by Snell's law:

$$\mu_{0n} = \sqrt{1 - (1-\mu_0^2)/m^2} \qquad (9)$$

For uniformly refractive media like snow on land, one can just set the refractive index $m_{re}$
equal to 1 for every layer. In this case, the Fresnel reflectance $R(-\mu_0,m)$ is 0 in equation
(7), the Fresenal transmittance $T(-\mu_0,m)$ is 1 in equation (8), and $\mu_{0n}$ equals to $\mu_0$: the
two source terms $S^a(\tau,\mu)$ and $S^b(\tau,\mu)$ become the same and equal to the source term of
homogenous media given in equation (2).

For 2-stream approximations of this method, analytical solutions of upward and
downward fluxes are coupled at each layer interface to generate 2N equations with 2N
unknown coefficients for any N-layer stratified column. The solutions of 2-stream
algorithms and boundary conditions for homogenous media are well documented
(Sections 8.4 and 8.10 of Thomas and Stamnes, 1999). Despite the extra source terms,
these 2N equations can also be organized into a tridiagonal matrix similar to the method
of Toon et al. (1989) used in SNICAR. Flexibility and speed therefore make this 2-stream
discrete-ordinate algorithm (hereafter, 2SD) a potentially good candidate for long-term
Earth system modeling. In this work, we only apply 2SD to snowpack and note that it can
be applied to any uniformly or non-uniformly refractive media like snow on land or sea
ice, with the Delta-transform implemented to fundamental optical variables (Equations 2
(a)-(c), Wiscombe and Warren, 1980).

2.4 16-stream DISORT
Besides the mathematical technique, the accuracy and speed of radiative transfer
algorithms depend on the number of angles used for flux estimation in the upward and
downward hemispheres. The algorithms used in SNICAR, Icepack, and 2SD use one
angle to represent upward flux and one angle to represent downward flux, hence they are
named 2-stream algorithm. Lee and Liou (2007) use two upward and two downward
streams. Jin and Stamnes (1994) documented the solutions for any even number of



streams. The speed of these models is slower than 2-stream models while their accuracy
is better. To quantify the accuracy of the three 2-stream algorithms for snow shortwave
simulations, we use the 16-stream DIScrete-Odinate Radiative Transfer model (DISORT)
as the benchmark model (http://lllab.phy.stevens.edu/disort/) (Stamnes et al., 1988).

**3. Input for radiative transfer models**
In this work, we focus on the performance of 2-stream algorithms for pure snow
simulations. The inputs for these three models are the same: single-scattering properties
(SSPs, i.e. single-scattering albedo $\varpi$, asymmetry factor $g$, extinction coefficient $\sigma_{ext}$) of
snow determined by snow grain radius r, snow depth, solar zenith angle $\theta$, solar incident
flux, and the albedo of underlying ground (assuming Lambertian reflectance of 0.25 for
all wavelengths). A Delta-transform is applied to fundamental input optical variables for
all simulations (Equations 2 (a)-(c), Wiscombe and Warren, 1980).

In snow, photon scattering occurs at the air-ice interface, and the absorption of photons
occurs within the ice crystal. The most important factor that determines snow shortwave
properties is the ratio of total surface area to total mass of snow grains, aka "the specific
surface area" (e.g. Matzl and Schneebeli, 2006, 2010). The specific surface area ($\beta$) can
be converted to a radiatively effective snow grain radius r:

$\beta = 3 / (r \varrho_{ice})$                                                      (10)

where $\rho_{ice}$ is the density of pure ice, 917 kg m$^{-3}$. Assuming the grains are spherical, the
SSPs of snow can thus be computed using Mie theory (Wiscombe, 1980) and ice optical
constants (Warren and Brandt, 2008). In nature, snow grains are not spherical, and many
studies have been carried out to quantify the accuracy of such spherical representations
(Grenfell and Warren, 1999; Neshyba et al., 2003; Grenfell et al., 2005). In recent years,
more research has been done to evaluate the impact of grain shape on snow shortwave
properties (Dang et al., 2016; He et al., 2017, 2018ab), and they show that non-spherical
snow grain shapes mainly alter the asymmetry factor. Dang et al., (2016) also point out
that the solar properties of a snowpack consisting of non-spherical ice grains can be
mimicked by a snowpack consisting of spherical grains with a smaller grain size by
factors up to 2.4. In this work, we still assume the snow grains are spherical, and this
assumption does not qualitatively alter our evaluation of the radiative transfer algorithms.



The input SSPs of snow grains are computed using Mie theory at fine spectral resolution
for a wide range of ice effective radius $r$ from 10 to 3000 μm that covers the possible
range of grain radius for snow on Earth (Flanner et al., 2007). The same spectral SSPs
were also used to derive the band-averaged SSPs of snow used in SNICAR. Note
Briegleb and Light (2007) refer to SSPs as inherent optical properties.

**4. Solar spectra used for the spectral integrations**
In climate modeling, snow albedo computation at fine spectral resolution is expensive
and unnecessary. Instead of computing spectrally resolved snow albedo as shown in
Figure 1, wider-band solar properties are more practical. For example, CESM and E3SM
aggregate the narrow RRTMG bands used for the atmospheric radiative transfer
simulation into visible (0.2 - 0.7 μm) and near-IR (0.7 - 5 μm) bands. The land model and
sea-ice model thus receive visible and near-IR fluxes as the upper boundary condition,
and return the corresponding visible and near-IR albedos to atmosphere model. In
practice, these bands are also partitioned into direct and diffuse components. Therefore, a
practical 2-stream algorithm should be able to simulate the direct visible, diffuse visible,
direct near-IR and diffuse near-IR albedos and absorptions of snow accurately.

The band albedo $\alpha$ is an irradiance-weighted average of the spectral albedo $\alpha(\lambda)$:

$$\alpha = \frac{\int_{\lambda 1}^{\lambda 2} \alpha(\lambda) F(\lambda) d\lambda}{\int_{\lambda 1}^{\lambda 2} F(\lambda) d\lambda} \qquad (11)$$

In this work, we use the spectral irradiance $F(\lambda)$ generated by the atmospheric DISORT-
based Shortwave Narrowband Model (SWNB2) (Zender et al., 1997; Zender, 1999) for
typical clear-sky and cloudy-sky conditions of mid-latitude winter as shown in Figure
1(a). The total clear-sky down-welling surface flux at different solar zenith angles are
also given in Figure 1(b).

**5. Model Evaluation**
5.1 Spectral albedo and reflected solar flux
The spectral reflectance of pure deep snow computed using 2-stream models and 16-
stream DISORT are shown in Figure 2. The snow grain radius is 100 μm - a typical grain
size for fresh new snow. For clear sky with direct beam source (left column), all three 2-
stream models show good accuracy at visible wavelengths (0.3 – 0.7 μm), and within this
band, the snow albedo is large and close to 1. As wavelength increases, the albedo
diminishes in the near-IR band. 2-stream models overestimate snow albedo at these
wavelengths, with maximum biases of 0.013 (SNICAR and CICE) and 0.023 (2SD)
within wavelength 1 - 1.7 μm. For cloudy-sky cases with diffuse upper boundary
conditions, CICE reproduces the snow albedo at all wavelengths with the smallest
absolute error (< 0.005), SNICAR and 2SD both overestimate the snow albedo with
maximum biases > 0.04 between 1.1-1.4 μm.
In both sky conditions, the errors of snow albedo are larger at near-IR wavelengths
ranging from 1.0-1.7 μm, while the solar incident flux peaks at 0.5 μm then decreases as
wavelength increases. The largest error in reflected flux is within the 0.7-1.5 μm band for
SNICAR and 2SD, as shown in the 3[rd] row of Figure 2. CICE overestimate the direct
snow albedo mostly at wavelengths larger than 1.5 μm where the error in reflected flux is
almost negligible.
5.2 Broadband albedo and reflected solar flux
Integrated over the visible and near-IR wavelengths, the error in band albedos computed
using 2-stream models for different cases are shown in Figure 3-6.
Figure 3 shows the error in direct band albedo for fixed snow grain radius of 100 μm with
different snow depth and solar zenith angles. As introduced in Section 2, SNICAR and
CICE both use delta-Eddington method to compute the visible albedo. They overestimate
the visible albedo for solar zenith angles smaller than 50° by up to 0.005, and
underestimate it for solar zenith angles larger than 50° by up to -0.01. 2SD produces
similar results for the visible band but at a larger solar zenith angle threshold of 75°. In
the near-IR band, SNICAR and 2SD overestimate the snow albedo for solar zenith angles
smaller than 70°, beyond this, the error in albedo increases by up to -0.1 as solar zenith
angle increases. CICE produces a similar error pattern with a smaller solar zenith angle
threshold at 60°. As snow ages, its average grain size increases. For typical old melting
snow of grain radius 1000 μm (Figure 4), 2-stream models produce similar errors of
direct albedo in all bands. For snow consisting of smaller grain size, 2-stream models
produce larger errors for visible albedo. Integrating over the entire solar band, the three 2-
stream models evaluated show similar error patterns for direct albedo.
For a fixed solar zenith angle of 60°, the error of direct albedo for different snow depth
and snow grain radii are shown in Figure 5. SNICAR and CICE underestimate the visible




albedo in most scenarios, while 2SD overestimates the visible albedo for a larger range of
grain radius and snow depth. All three 2-stream models tend to overestimate the near-IR
albedo except for shallow snow with large grain radius; the error of 2SD is one order of
magnitude larger than that of SNICAR and CICE.

Figure 6 is similar to Figure 5, but shows the diffuse snow albedo. In the visible band,
SNICAR and CICE generate similar errors in that they both underestimate the albedo as
snow grain size increases and snow depth decreases. 2SD overestimates the albedo with
maximum error of around 0.015. In the near-IR, 2-stream models tend to overestimate
snow albedo, while the magnitude of biases produced by SNICAR and 2SD are one order
larger than that of CICE with the maximum error of 0.035 generated by SNICAR. As a
result, the all-wave diffuse albedos computed using CICE are more accurate than those
computed using SNICAR and 2SD.

Figures 7, 8 and 9 show the errors in reflected shortwave flux caused by snow albedo
errors seen in Figures 3, 4, and 6. In general, 2-stream models produce larger errors in
reflected direct near-IR flux (Figure 7 and 8), especially with the 2SD model: the
maximum overestimate of reflected near-IR flux is 6-8 $Wm^{-2}$ for deep melting snow with
solar zenith angle < 30°. Errors in reflected direct visible flux are smaller (mostly within
±1 $Wm^{-2}$)for all models in most scenarios, and become larger (mostly within ±3 $Wm^{-2}$) as
snow grain size increases to 1000 μm if computed using 2SD. As shown in Figure 9, for
diffuse flux with solar zenith angle of 60° at TOA, SNICAR and CICE generate small
errors in reflected visible flux (mostly within ±1 $Wm^{-2}$), while 2SD always overestimates
reflected visible flux by up to 5 $Wm^{-2}$. In the near-IR, SNICAR and 2SD overestimate
reflected flux by as much as 10-12 $Wm^{-2}$; the error in reflected near-IR flux produced by
CICE is much smaller, mostly within ±1 $Wm^{-2}$.

In general, CICE produces the most accurate albedo and thus reflected flux for both
direct and diffuse components. SNICAR is similar to CICE for its accuracy of direct
albedo and flux, yet generates large error for diffuse component. 2SD tends to
overestimate snow albedo and reflected flux in both direct and diffuse components and
shows the largest errors among three 2-stream models. Note that the final errors of snow
albedo and reflected solar flux are the weighted sum of direct and diffuse components,
and their weights are largely determined by cloud cover fraction (e.g. Figure 6, Dang et
al., 2017), which we do not address explicitly in this paper.




## 5.3 Band absorption of solar flux

Figure 10 shows absorption profiles of shortwave flux computed using the 16-stream
DISORT model, with errors in absorbed fractional solar flux computed using 2-stream
models. The snowpack is 10-cm deep, and is divided into 5 layers, each 2-cm thick. The
snow grain radius is set to 100 μm. The figure shows fractional absorption for snow
layers 1-4 and the underlying ground with albedo of 0.25.

As shown in the first column of Figure 10, for new snow with radius of 100 μm, most
solar absorption occurs in the top 2-cm snow layer, where roughly 10% and 15% of
diffuse and direct near-IR flux are absorbed and dominate the solar absorption within
snowpack. In the second layer (2-4 cm), the absorption of solar flux is less than 1% and
gradually decreases within the interior layers. The underlying ground absorbs roughly 2%
of solar flux, mostly visible flux that penetrates the snowpack more efficiently. As snow
ages and snow grain grows, photons penetrate deeper into the snowpack. For typical old
melting snow with radius of 1000 μm, most solar absorption still occurs in the top 2-cm
snow layer, where roughly 20% and 14% of diffuse and direct near-IR flux are absorbed.
The second snow layer (2-4 cm) absorbs more near-IR solar flux by roughly 2%. More
photons are able to penetrate through the snowpack, and results in a high fractionally
absorption by the underlying ground, especially for visible band. As snow depth increase,
the ground absorption will decrease for both snow radii.

Comparing to 16-stream DISORT, 2-stream models underestimate (overestimate) the
column solar absorptions for new (old) snow, especially for the surface snow layer and
ground layer. Overall, CICE gives the most accurate absorption profiles among three 2-
stream models, especially for new snow.

## 6. Correction for direct albedo for large solar zenith angles

It has been pointed out in previous studies that the 2-stream approximations become poor
as solar zenith angle approaches 90° (e.g. Wiscombe 1997, Warren 1982). As shown in
Figures 3 and 4, all three 2-stream models underestimate the direct snow albedo for large
solar zenith angles. In the visible band, when snow grain size is small, the error in direct
albedo is almost negligible (Figure 3); while as snow ages and snow grains become
larger, the error increases yet still remains low if the snow is deep (Figure 4). In the near-
IR, the biases of albedo are also larger for larger snow grain radii. For a given snow size,





the magnitudes of such biases are almost independent of snow depth, and mainly
determined by the solar zenith angle. In general, the errors of all-wave direct albedo are
mostly contributed by the errors of near-IR albedo, especially for optically thick
snowpacks (i.e., semi-infinite), because the errors of direct albedo in the visible are
negligible compared with those in the near-IR. To improve the performance of 2-stream
algorithms, we develop a parameterization that corrects the underestimated near-IR snow
albedo at large zenith angles.

Figure 11 shows the direct near-IR albedo and fractional absorption of a 2-meter thick
snowpacks consisting of grains with radius 100 μm and 1000 μm, computed using 2-
stream algorithms and 16-stream DISORT. For solar zenith angles > 75°, 2-stream
models underestimate snow albedo and overestimate solar absorption within snowpack,
mostly in the top 2-cm of snow. We define and compute $R_{75+}$ as the ratio of direct semi-
infinite near-IR albedo computed using 16-stream DISORT ($\alpha_{16\text{-}DISORT}$) to that computed
using CICE ($\alpha_{CICE}$). This ratio is shown in Figure 11 (c) and can be parameterized as a
function of snow grain radius ($r$, unit in meter) and the cosine of incident solar zenith
angle ($\mu_0$), as shown in Figure 11(c):

$R_{75+} = \dfrac{\alpha_{16\text{-}DISORT}}{\alpha_{CICE}} = c_1(\mu_0)log_{10}(r) + c_0(\mu_0)$          (12)

where coefficients $c_1$ and $c_0$ are polynomial functions of $\mu_0$, as shown in Figure 11(d):

$c_1(\mu_0) = 1.304\mu_0^2 - 0.631\mu_0 + 0.086$                    (13a)
$c_0(\mu_0) = 6.807\mu_0^2 - 3.338\mu_0 + 1.467$                    (13b)

Since 2-stream models always underestimate snow albedo, $R_{75+}$ always exceeds 1 (Figure
11c). We can then adjust the direct near-IR snow albedo ($\alpha_{CICE}$) and direct near-IR solar
absorption ($Fabs_{CICE}$) by snow computed using CICE with ratio R$_{75+}$:

$\alpha_{CICE}^{adjust} = R_{75+}\alpha_{CICE}$                    (14a)

$Fabs_{CICE}^{adjust} = Fabs_{CICE} - (R_{75+} - 1) * \alpha_{CICE} * F_{nir}$          (14b)





where $F_{nir}$ is the direct near-IR flux. This adjustment reduces the error of near-IR albedo from negative 2-10% to within ± 0.5% for solar zenith angles larger than 75°, and for grain radii ranging from 30-1500 μm (Figure 12). Errors in broadband direct albedo are therefore also reduced to < 0.01. The direct near-IR flux absorbed by the snowpack decreases after applying this adjustment. In practice, if snow is divided into multiple layers, we assume all decreased near-IR absorption (2$^{nd}$ term on the right hand side, equation 14b) is confined within the top layer. This assumption is fairly accurate for the near-IR band, since most direct IR absorption occurs at the very surface of snowpack (Figures 10 and 11).

It is important to note that although the errors of direct near-IR albedos are large for large solar zenith angles, the absolute error in reflected shortwave flux is small (Figures 7 and 8) as the down-welling solar flux reaches snowpack decreases as solar zenith angle increases (Figures 1(b)). However, such small biases in flux can be important at high latitudes where the solar zenith angle remains large for many days in late winter and early spring. We have implemented this parameterization in MPAS-seaice to quantify its impact on polar climate, though these experiments are beyond the scope of the present paper.

## 7. Implementation of snow radiative transfer model in Earth system models

ESMs often use broader band-averaged SSPs of snow and aerosols for computational efficiency, rather than using brute-force integration of spectral solar properties across narrower bands (per equation 11). Besides using different radiative transfer approximations, SNICAR and CICE also adopt different methods to derive the band-averaged SSPs of snow for different band schemes.

In SNICAR, snow solar properties are computed for 5 bands: one visible band (0.3 - 0.7μm), and four near-IR bands (0.7 - 1 μm, 1 – 1.2 μm, 1.2 – 1.5 μm, and 1.5 – 5 μm). The solar properties of four subdivided near-IR bands are combined by fixed ratios to compute the direct/diffuse near-IR snow properties. These two sets of ratios are derived offline based on the incident solar spectra of typical of mid-latitude winter for clear and cloudy-sky conditions clear sky and cloudy sky, respectively (Figure 1(a)).

The band-averaged SSPs of snow grains are computed following the Chandrasekhar Mean approach (Thomas and Stamnes, 1999, their Equation 9.27; Flanner et al., 2007).



Specifically, spectral SSPs of snow grains are weighted into bands according to surface
incident solar flux typical of mid-latitude winter for clear and cloudy sky conditions. In
addition, the single-scattering albedo $\varpi(\lambda)$ of ice grains are also weighted by the
hemispheric albedo $\alpha(\lambda)$ of an optically thick snowpack:

$\varpi(\bar{\lambda}) = \dfrac{\int_{\lambda_1}^{\lambda_2} \varpi(\lambda) F(\lambda) \alpha(\lambda) d\lambda}{\int_{\lambda_1}^{\lambda_2} F(\lambda) \alpha(\lambda) d\lambda}$    (15a)
$g(\bar{\lambda}) = \dfrac{\int_{\lambda_1}^{\lambda_2} g(\lambda) F(\lambda) d\lambda}{\int_{\lambda_1}^{\lambda_2} F(\lambda) \alpha(\lambda) d\lambda}$    (15b)
$\sigma_{ext}(\bar{\lambda}) = \dfrac{\int_{\lambda_1}^{\lambda_2} \sigma_{ext}(\lambda) F(\lambda) d\lambda}{\int_{\lambda_1}^{\lambda_2} F(\lambda) \alpha(\lambda) d\lambda}$    (15c)

Two sets of snow band-averaged SSPs are generated for all grain radii, suitable for direct
and diffuse light, respectively. For each modeling step and band, SNICAR is called twice
to compute the direct and diffuse snow solar properties.

In CICE, the snow-covered sea ice properties are computed for 3 bands: one visible band
(0.3 – 07 μm), and two near-IR bands (0.7 – 1.19 μm and 1.19 – 5 μm). The solar
proprieties of these two near-IR bands are combined using ratios $w_{nir1}$ and $w_{nir2}$ for 0.7-1
.19 μm and 1.19-5 μm, depending on the fraction of direct near-IR flux $f_{nidr}$:

$w_{nir1} = 0.67 + 0.11 * (1 - f_{nidr})$    (16a)
$w_{nir2} = 1 - w_{nir1}$    (16b)

The band SSPs of snow are derived by integrating the spectral SSPs and the spectral
surface solar irradiance measured in the Arctic under mostly clear sky.

$\varpi(\bar{\lambda}) = \int_{\lambda_1}^{\lambda_2} \varpi(\lambda) F(\lambda) d\lambda$    (17a)
$g(\bar{\lambda}) = \int_{\lambda_1}^{\lambda_2} g(\lambda) F(\lambda) d\lambda$    (17b)
$\sigma_{ext}(\bar{\lambda}) = \int_{\lambda_1}^{\lambda_2} \sigma_{ext}(\lambda) F(\lambda) d\lambda$    (17c)





In addition, the band-averaged single-scattering albedo $\varpi(\bar{\lambda})$ is also increased to $\varpi(\bar{\lambda})'$
until the band albedo computed using averaged SSPs matches the band albedo $\bar{\alpha}$ within
0.0001, where $\bar{\alpha}$ is:

$\bar{\alpha} = \int_{\lambda_1}^{\lambda_2} \alpha(\lambda)F(\lambda)d\lambda$                          (18)

CICE adopts this single set of band SSPs for both direct and diffuse computations. In
practice, the physical snow grain radius $r$ is adjusted to a radiatively equivalent radius $r_{eqv}$
based on the fraction of direct flux in the near-IR band ($f_{nidr}$):

$r_{eqv} = (f_{nidr} + 0.8(1 - f_{nidr}))r$                          (19)

This $r_{eqv}$ and the corresponding snow SSPs are then used in the radiative transfer
calculation. The computed direct and diffuse solar properties alone are less accurate,
while the combined all-sky broadband solar properties agree with SNICAR (Briegleb and
Light, 2007). As a result, for each modeling step and band, CICE radiative transfer
subroutine is called only once to compute both the direct and diffuse snow solar
properties simultaneously.

SNICAR and CICE also use different approaches to avoid numerical singularities. In
SNICAR, singularities occur when the denominator of term $C_n^{\pm}$ in equation (3) equals to
zero (i.e., $\gamma^2 - 1/\mu_0^2 = 0$), where $\gamma$ is determined by the approximation method and SSPs
of snow, and $\mu_0$ is the cosine of the solar zenith angle (Equations 23 and 24, Toon et al.,
1989). When such a singularity is detected, SNICAR will shift $\mu_0$ by + 0.02 or -0.02 to
obtain physically realistic radiative properties. In the CICE algorithm, singularities arise
only when $\mu_0 = 0$ (Equation 4). Therefore, in practice, for $\mu_0 < 0.01$, CICE computes the
sea-ice solar properties for $\mu_0 = 0.01$ to avoid unphysical results.

**8. Discussion: a unified radiative transfer model for snow, sea ice, and land ice.**
Based on the inter-comparison of three 2-stream algorithms and their implementations in
ESMs,    we    formulated    the    following    surface    shortwave    radiative    transfer



recommendations for an accurate, fast, and consistent treatment for snow on land, land
ice, and sea ice in ESMs:

First, the 2-stream delta-Eddington adding-doubling algorithm by Briegleb and Light
(2007) is unsurpassed as a radiative transfer core. The evaluation in Section 5 shows that
this algorithm produces the least error for snow albedo and solar absorption within
snowpack, especially under overcast sky. This algorithm applies well to both uniformly
refractive media such as snow on land, and to non-uniformly refractive media, such as
bare/snow-covered/ponded sea ice and bare/snow-covered land ice. Numerical
singularities occur only rarely (when $\mu_0 = 0$) and are easily avoided in model
implementations. Among the three 2-stream algorithms discussed here, the CICE
radiative core is also the most efficient one as it takes only ~2/3 of the time of SNICAR
and 2SD to compute solar properties of multi-layer snowpacks.

Second, any 2-stream cryospheric radiative transfer model can incorporate the
parameterization described in Section 6 to adjust the low bias of direct near-IR snow
albedo and high bias of direct near-IR solar absorption in snow, for solar zenith angles
larger than 75°. These biases are persistent across all 2-stream algorithms discussed in
this work, and should be corrected for snow-covered surfaces. Alternatively, adopting a
4-stream approximation would reduce or eliminate such biases, though at considerable
expense in computational efficiency.

Third, a cryospheric radiative transfer model should prefer physically based
parameterizations that are extensible and convergent (e.g., with increasing spectral
resolution) for the band-averaged SSPs and size distribution of snow. Although the
treatments used in SNICAR and CICE are both practical since they both reproduce the
narrowband solar properties with carefully derived band-averaged inputs as discussed in
Section 7, the snow treatment used in SNICAR is more physically based and reproducible
since it does not rely on subjective adjustment and empirical coefficients as used in
CICE. Specifically, the empirical adjustment to snow grain radius implemented in CICE
may not always produce compensating errors. For example, in snow containing light-
absorbing impurities such adjustment may also lead to biases in aerosol absorption since
the albedo reduction caused by light-absorbing particles does not linearly depend on
snow grain radius (Dang et al., 2015). For further model development incorporating non-
spherical snow grain shapes (Dang et al., 2016; He et al., 2018ab), such adjustment on
grain radius may fail as well. Moreover, SNICAR computes the snow properties for four



near-IR bands, which helps capture the spectral variation of albedo (Figure 2) and
therefore better represents near-IR solar properties. It is also worth noting that unlike the
radiative core of CICE, SNICAR is actively maintained with numerous modifications and
updates in the past decade (e.g. Flanner et al., 2012; He et al., 2018b). Snow radiative
treatments that follow SNICAR conventions for SSPs may take advantage of these
updates. Note that any radiative core that follows SNICAR SSP conventions must be
called twice to compute diffuse and direct solar properties, respectively.
Fourth, a surface cryospheric radiative transfer model should flexibly accommodate
coupled simulations with distinct atmospheric and surface spectral grids. Both the 5-band
scheme used in SNICAR and the 3-band scheme used in CICE separate the visible from
near-IR spectrum at 0.7 μm. This boundary aligns with the Community Atmospheric
Model's original radiation bands (CAM; Neale et al., 2012), though not with the widely
used Rapid Radiative Transfer Model (RRTMG; Iacono et al., 2008) which places 0.7 μm
squarely in the middle of a spectral band. A mismatch in spectral boundaries between
atmospheric and surface radiative transfer schemes can require an ESM to unphysically
apportion energy from the straddled spectral bin when coupling fluxes between surface
and atmosphere. The spectral grids of surface and atmosphere radiation need not be
identical so long as the coarser grid shares spectral boundaries with the finer grid. In
practice maintaining a portable cryospheric radiative module such as SNICAR requires a
complex offline toolchain (Mie solver, spectral refractive indices for air, water, ice, and
aerosols, spectral solar insolation for clear and cloudy skies) to compute, integrate, and
rebin SSPs. Aligned spectral boundaries between surface and atmospheric would simplify
the development of efficient and accurate radiative transfer for the coupled Earth system.
Last, it is important to note that, although we only examine the performance of the CICE
adding-doubling algorithm for pure snow in this work, this algorithm can be applied to
the surface solar calculation of all cryospheric components with or without light-
absorbing particles present. First, Briegleb and Light (2007) proved its accuracy for
simulating ponded/bare sea-ice solar properties against observations and a Monte Carlo
radiation model. Second, In CESM and E3SM, the radiative transfer simulation of snow
on land ice is carried out by SNICAR with prescribed land ice albedo. Adopting the
CICE adding-doubling core in SNICAR will permit these ESMs to couple the snow and
land ice as a non-uniformly refractive column for more accurate solar computations since
bare/snow-covered/ponded land ice is physically similar to bare/snow-covered/ponded
sea ice, and the latter is already treated well by CICE radiative transfer core. Third,



adding light-absorbing particles in snow will not change our results qualitatively. Both
CICE and SNICAR simulate the impact of light-absorbing particles (black carbon and
dust) on snow and/or sea ice using self-consistent particle SSPs that follow the SNICAR
convention. The adoption of CICE radiative transfer algorithm in SNICAR, and the
implementation of SNICAR snow SSPs in CICE will enable a consistent simulation on
the radiative effects of light-absorbing particles in the cryosphere across ESM
components.
In summary, this inter-comparison and evaluation has shown multiple ways that the solar
properties of cryospheric surfaces can be improved in the current generation of ESMs.
We have adopted these recommendations in a hybrid model SNICAR-AD, implemented
in MPAS-seaice and E3SM Land Model (ELM), to examine the response of climate to
this improved and unified cryospheric surface radiation treatment in future E3SM studies.
**9. Conclusions**
In this work, we aim to improve and unify the solar radiative transfer calculations for
snow on land and snow on sea ice in ESMs by evaluating the following 2-stream
radiative transfer algorithms:  the 2-stream delta-Eddington adding-doubling algorithm
implemented in sea-ice model Icepack/CICE/MPAS-seaice, the 2-stream delta-Eddington
and 2-stream delta-Hemispheric-Mean algorithms implemented in snow model SNICAR,
and a 2-stream delta-Discrete-Ordinate algorithm. Among these three models, the 2-
stream delta-Eddington adding-doubling algorithm produces the most accurate snow
albedo and solar absorption (Section 5). All 2-stream models underestimate near-IR snow
albedo and overestimate near-IR absorption when solar zenith angles are larger than 75°,
which can be adjusted by a parameterization we developed (Section 6). We compared the
implementations of radiative transfer cores in SNICAR and CICE (Section 7) and
recommended a consistent shortwave radiative treatment for snow-covered surfaces
across ESMs (Section 8). Improved treatment of surface cryospheric radiative properties
in the thermal infrared has recently been shown to remediate significant climate
simulation biases in Polar Regions (Huang et al., 2018). It is hoped that adoption of
improved and consistent treatments of solar radiative properties for snow-covered
surfaces as described in this study (i.e. the hybrid model SNICAR-AD) will further
remediate simulation biases in Polar Regions.





**Data availability.** The data and models are available upon request to Cheng Dang
(cdang5@uci.edu). SNICAR and CICE radiative transfer core can be found at
https://github.com/E3SM-Project/E3SM.
**Competing interests.** The authors declare that they have no conflict of interest.
**Acknowledgements.** The authors thank Prof. Stephen G. Warren and Prof. Qiang Fu for
helpful discussions on radiative transfer algorithms. The authors thank Dr. Adrian Turner
for instructions on installing and running MPAS-seaice. This research was supported as
part of the Energy Exascale Earth System Model (E3SM) project, funded by the U.S.
Department of Energy, Office of Science, Office of Biological and Environmental
Research, with funding number DOE BER ESM (DE-SC0012998).

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



Figure 1. Spectral and total down-welling solar flux at surface computed using SWNB2
for (a) standard clear-sky and cloudy-sky atmospheric profiles of mid-latitude winter
assuming solar zenith angle is 60° at the top of atmosphere, and for (b) standard clear sky
profiles of mid-latitude and sub-Arctic winter with different incident solar zenith angles.

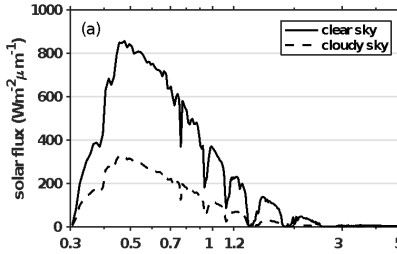
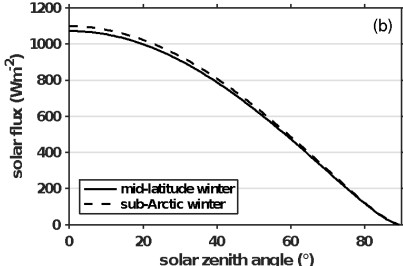




Figure 2. Spectral albedo of pure snow computed using 16-stream DISORT, SNICAR,
CICE, and 2SD models, for clear-sky (direct beam at solar zenith angle 60°) and cloudy-
sky conditions in the left and right panels, respectively. The top panels show spectral
albedo. The middle panels show the difference ($\delta\alpha = \alpha_2 - \alpha_{16}$) in spectral albedos
computed using 2-stream model ($\alpha_2$) and 16-stream DISORT ($\alpha_{16}$). The bottom panels
show the different of reflected spectral flux given $\delta\alpha$. The snowpack is set to semi-
infinite deep with grain radius of 100 μm.

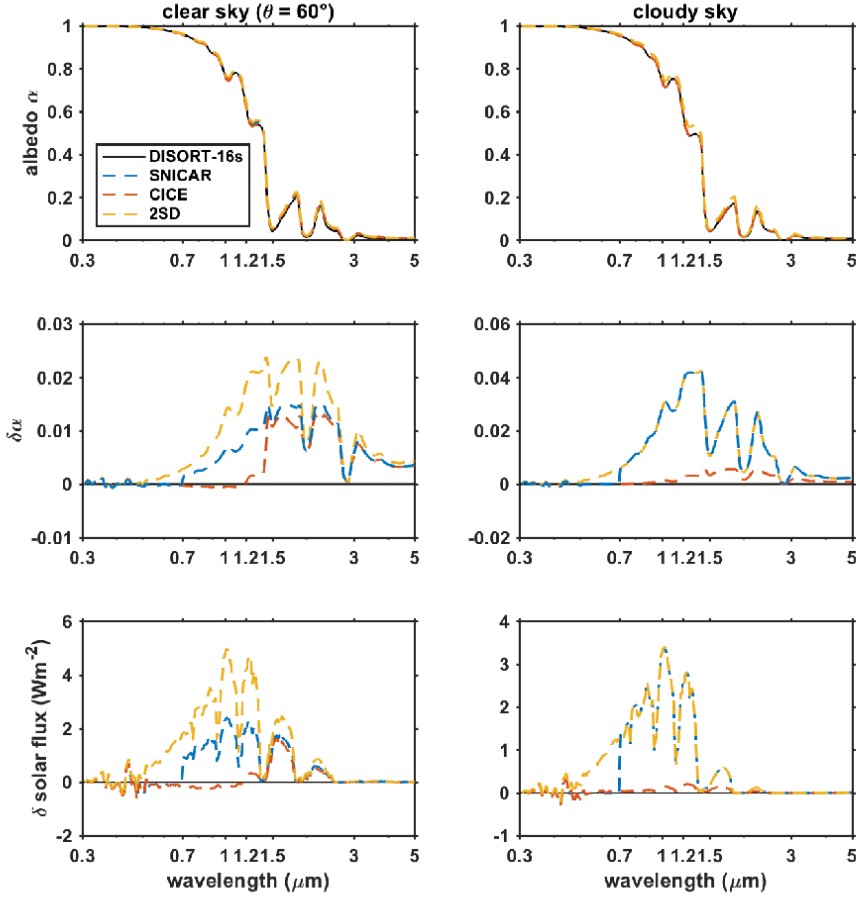




Figure 3. The difference in direct snow albedo ($\delta\alpha = \alpha_2 - \alpha_{16}$) computed using 2-stream
models ($\alpha_2$) and using 16-stream DISORT model ($\alpha_{16}$), for various snow depths and solar
zenith angles, with snow gran radius of 100 μm. From the top to the bottom rows are
results of 2-stream models SNICAR, CICE, and 2SD. From the left to the right columns
are albedo differences of all-wave, visible, near-IR bands.

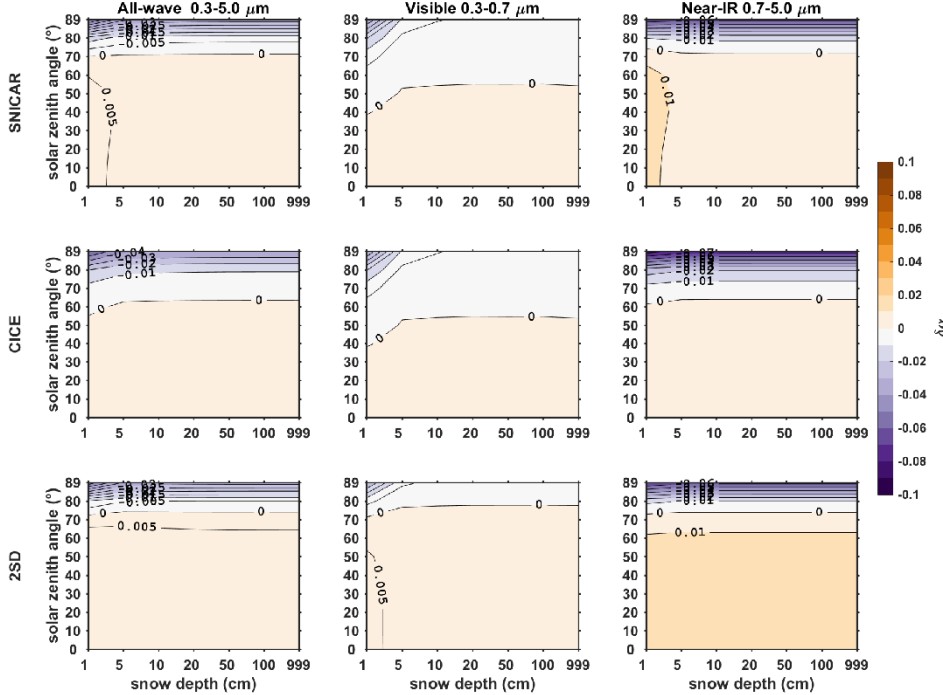




Figure 4. The same to Figure 3, but for snow grain radius of 1000 μm.

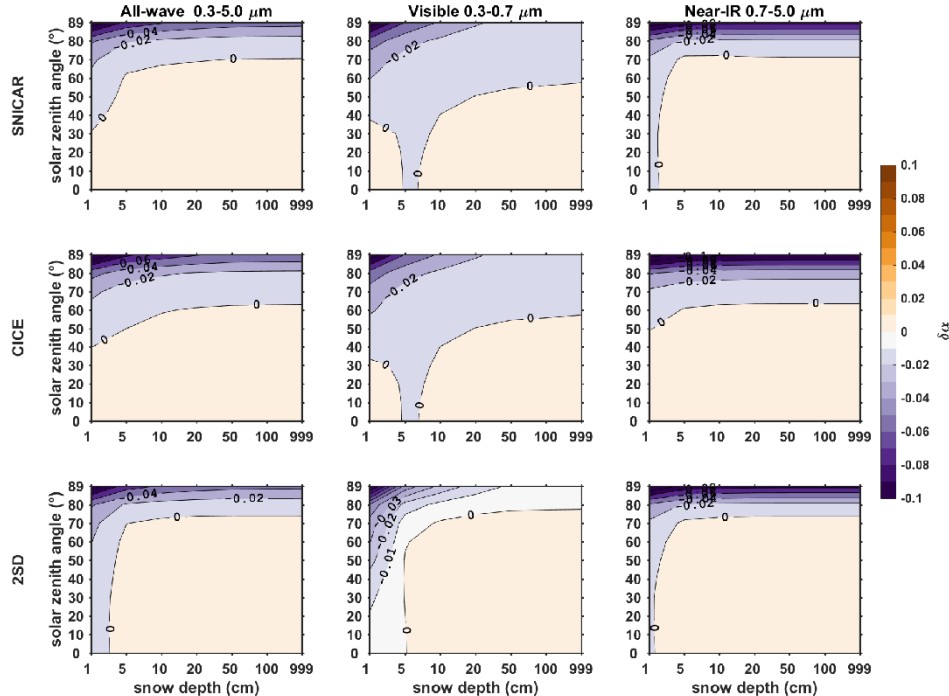






Figure 5. The same to Figure 3, but for fixed solar zenith angle of 60° and different snow
grain radii.

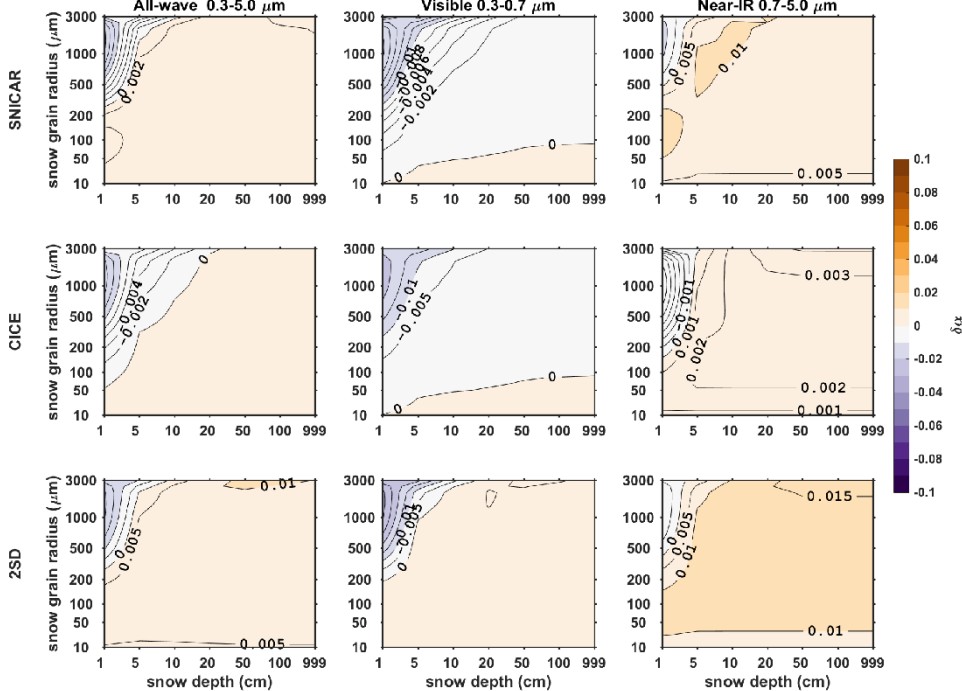




Figure 6. The same to Figure 5, but for diffuse snow albedo with different snow grain
radii.

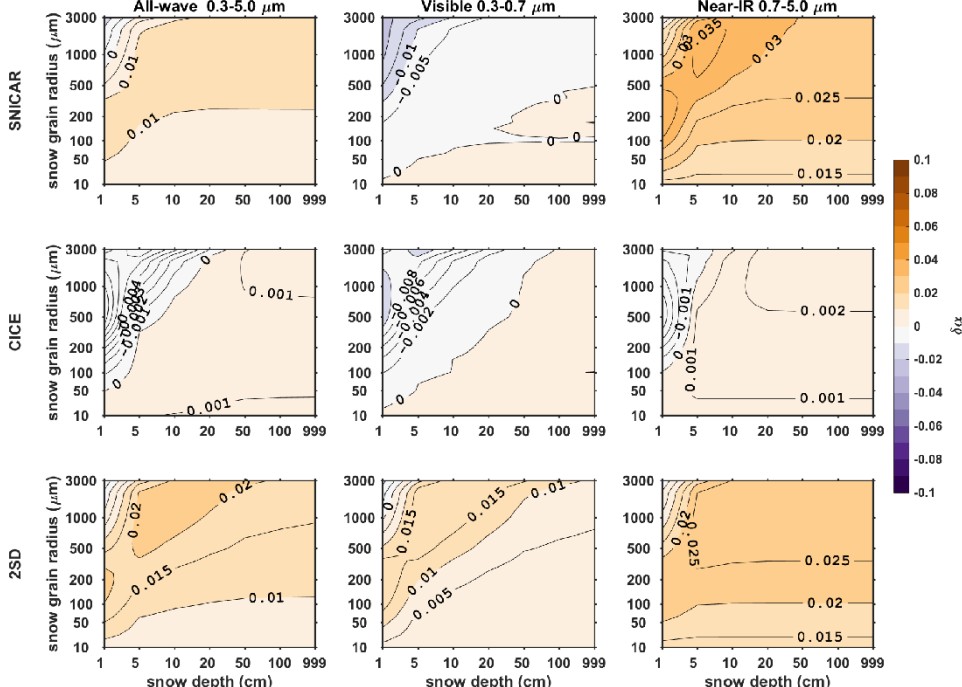




Figure 7. Error in reflected direct solar flux given albedo errors shown in Figure 3.

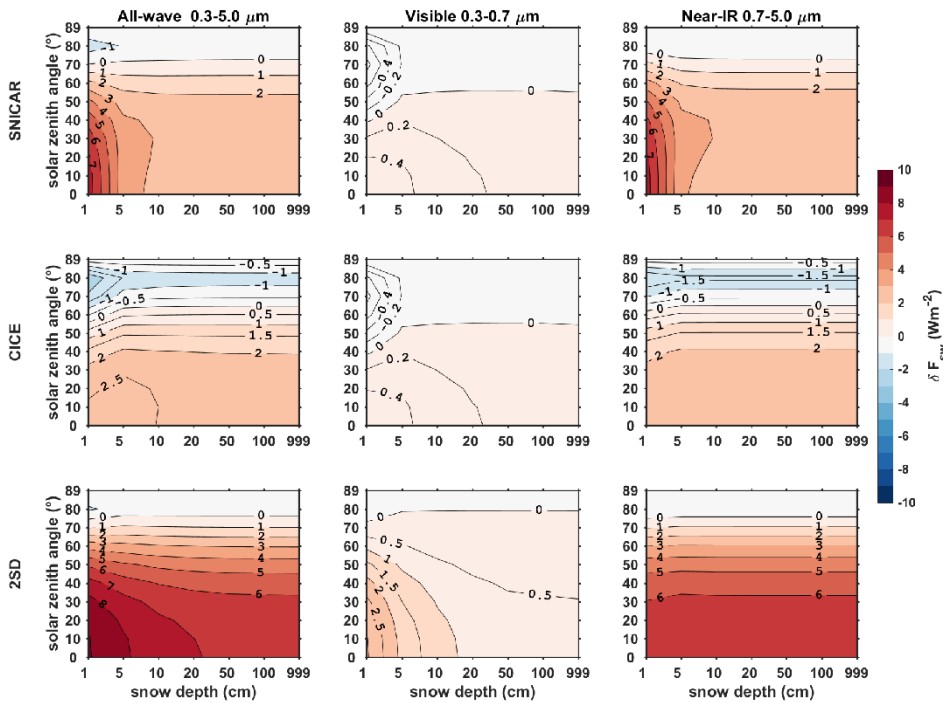




Figure 8. Error in reflected direct solar flux given albedo errors shown in Figure 4.

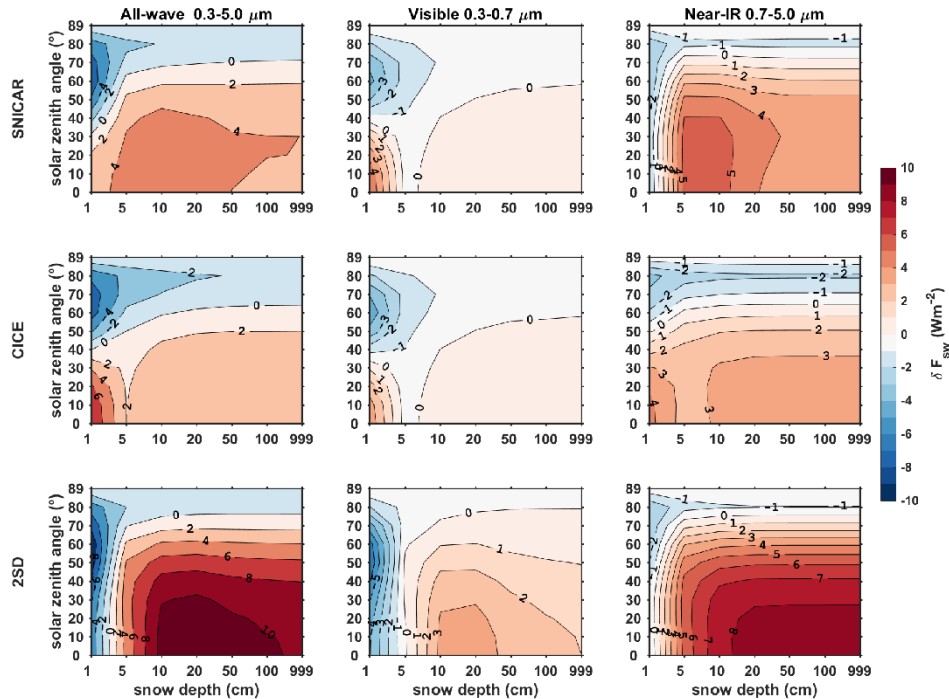




Figure 9. Error in reflected diffuse solar flux given albedo errors shown in Figure 6.

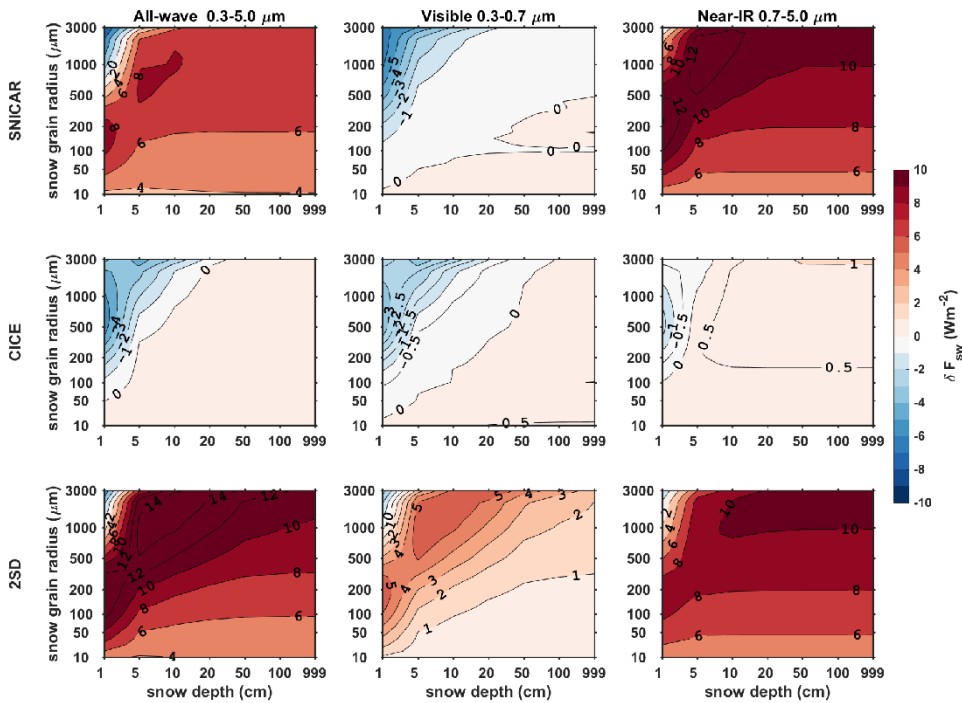




Figure 10. Comparison of light-absorption profiles derived from 2-stream models and 16-
stream DISORT. The left-most column show fractional band absorptions computed using
16-stream DISORT. The right three panels show the errors of all-wave, visible, and near-
IR fractional absorptions calculated using 2-stream models. The top and bottom panels
are for clear-sky and cloudy-sky conditions (solar zenith angle of 60°), respectively.  The
snowpack is 10 cm deep, and is divided evenly into five 2-cm thick layers, for new snow
(r = 100 μm) and old snow (r = 1000 μm). The layers 1-4 represent the top four snow
layers (top 8 cm), and layer 5 represents underlying ground with albedo of 0.25.

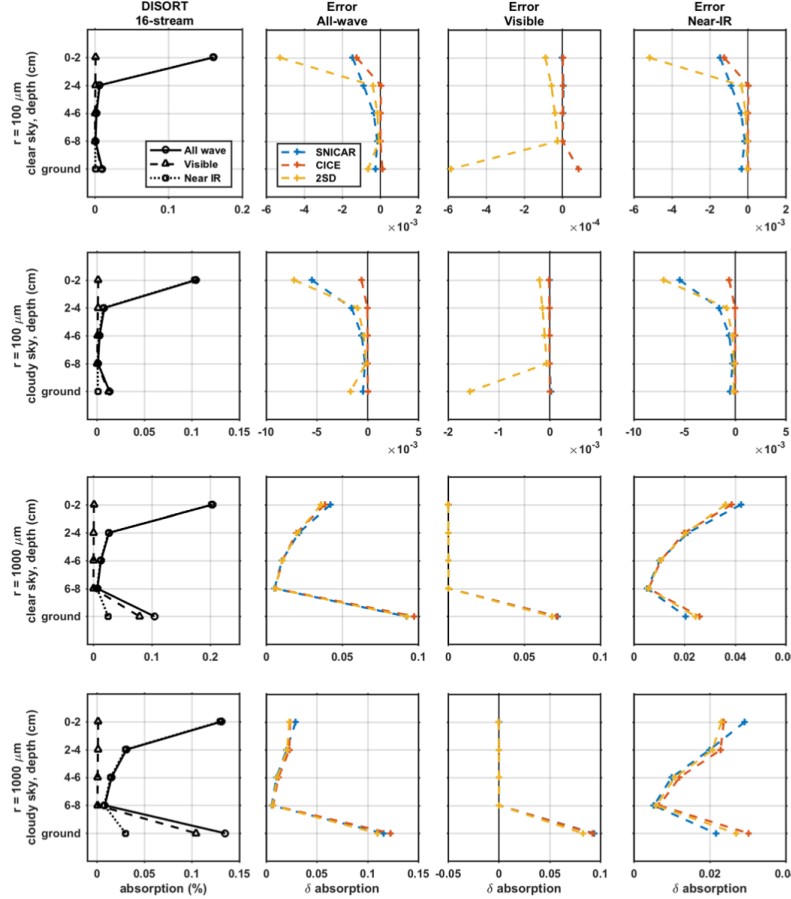






Figure 11. (a) Direct near-IR snow albedo and (b) near-IR fractional absorption by top 2-
cm snow of a 2-m thick snowpack, for solar zenith angles larger than 70° and snow grain
radii of 100 μm and 1000 μm. (c) The ratios of near-IR albedo computed using CICE to
that computed using 16-stream DISORT for different solar zenith angles. These ratios are
parameterized as liner functions of the logarithmic of snow grain radius. The slopes and
y-intercepts are shown in (d). The black dashed curves in figures (c) and (d) are fitting
values computed using parameterization discussed in Section 5.

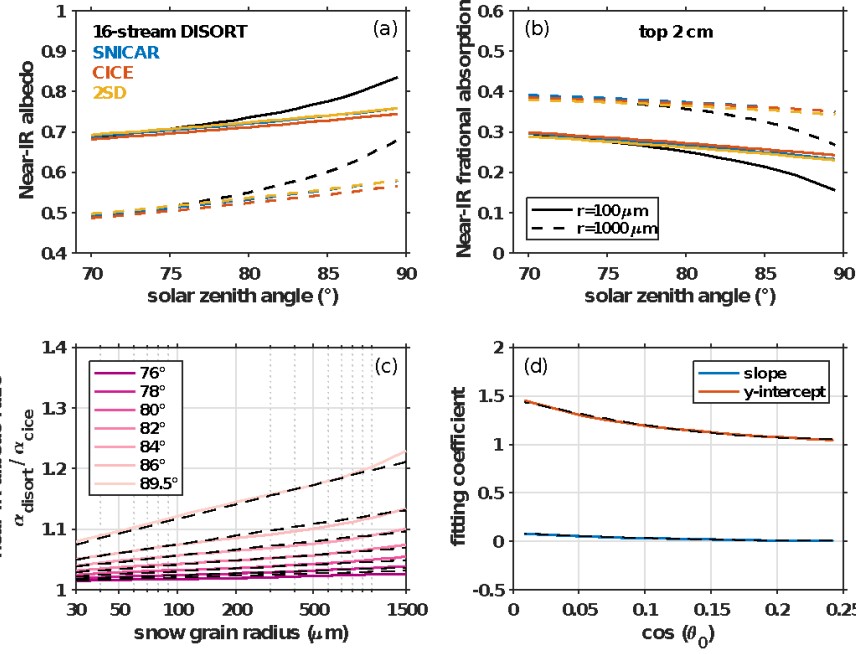






Figure 12. Error in semi-infinite snow albedo computed using CICE before (top row) and
after (bottom row) incorporating corrections for near-IR albedo, for different solar zenith
angles and snow grain radii.

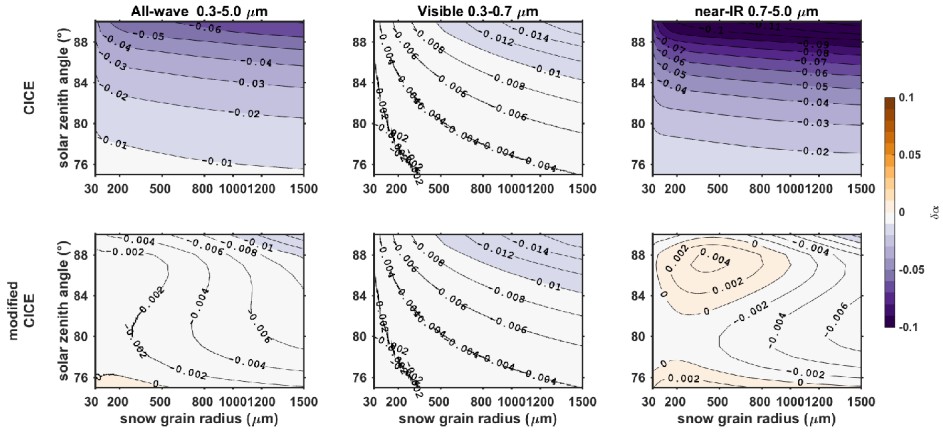




Table 1. Two-stream radiative transfer algorithms evaluated in this work, including
algorithms that are currently implemented in Earth System Model CESM and E3SM.

| ESM Component | Land | Sea Ice | |
|---|---|---|---|
| Model | SNICAR | CICE/MPAS-seaice | 2SD |
| Radiative transfer approximation | 2-stream δ-Eddington (visible) δ-Hemispheric-mean (near-IR) | 2-stream δ-Eddington | 2-stream δ-Discrete-ordinate |
| Treatment for multi-layered media | matrix inversion | adding-doubling | matrix inversion |
| Fresnel reflection/refraction | no | yes | yes |
| Number of bands implemented in ESMs | 5 bands (1 visible, 4 near-IR) | 3 bands (1 visible, 2 near-IR) | |
| Applies to | snow | bare/ponded/snow-covered sea ice, and snow | bare/ponded/snow-covered sea ice, and snow |
