# Peer review of "Inter-comparison and improvement of two-stream shortwave radiative transfer models in ESMs for a unified treatment of cryospheric surfaces"

_The Cryosphere, 2019_

## Referee Comment (RC1) · Anonymous Referee #1 · 14 Mar 2019

This paper aims to unify the treatment of the optical properties of snow and ice in ESMs, because, historically, there have often been different albedo schemes in one and the same ESM, applied to seasonal snow, snow on sea ice, and snow on land ice masses like ice sheets. In particular, the authors focus on SNICAR (used for land ice and land masses), and Icepack/CICE (for sea ice). All radiative transfer schemes in ESMs are two-stream approximations. They assess the accuracy of the 2-stream approximations of different flavors against DISORT in a 16-stream configuration and consider this as the benchmark.

[Figure]

...

I have few comments on the research itself. This is solid and has been carefully designed, and conducted.

I do have critical remarks on the presentation of the results. They are surprisingly unclear in a few key parts, to the extent that I am unsure what the authors have actually done and what they haven't done.

The problem lies in the fact that there is (1) a correction for high SZA > 75 degrees, and (2) mention of a hybrid model SNICAR-AD. To me it is unclear whether these are the same thing or not. The high-SZA correction seems to be carried out on CICE rather than SNICAR. The authors suggest that a correction (equations 13 a and b) can be conducted for any 2-stream approximation, so also for SNICAR.

So what is the situation after this paper? Do the authors now have 1 model for all snow and ice surfaces? Or did they present a correction for CICE only? Or also for SNICAR? And is this then SNICAR-AD? And what are the recommendation in section 8 about? Is this for future work? Or are these points that have been taken into account while a unified model framework was developed?

All in all, there are two possibilities: (1) either the authors have forgotten to mention that the correction for SZA > 75 degrees is actually called SNICAR-AD. In that case, Figure 12 needs different figure axis labels and the text needs to clarify that this correction is called SNICAR-AD in a few places. In this case, I would suggest to move section 8 forward between the current sections 5 and 6, so as to present first the requirements for a unified model, and then the actual unified model.

Or, (2) the correction for SZA > 75 degrees is not the same as SNICAR-AD, but rather an intermediate step in the unification of Icepack and SNICAR. In that case, the paper is incomplete. Results from the to-be-developed SNICAR-AD need to be incorporated here. If this would be paper 1 presenting the SZA-correction, and in a future paper 2 we will see SNICAR-AD fully developed, then I recommend that this paper is postponed to when SNICAR-AD is finished.

I recommend to the editor to inquire with the authors which of these possible situations is the case, and then reach a decision.

---

## Referee Comment (RC2) · David Bailey (Referee) · 18 Mar 2019

I have read the manuscript: "Inter-comparison and improvement of 2-stream shortwave radiative transfer models for a unified treatment of cryospheric surfaces in ESMs" by Dang et al. Overall, the article is interesting and may be of interest to the readers of the Cryosphere. I believe it needs some substantial revision before it would be acceptable however. There are a lot of model acronyms thrown around here and the text should be expanded to help clarify in some sections. Also, I feel it is lacking a bit in motivation. I understand from the title, the idea is to unify the radiative transfer schemes for snow

on land, ice sheets, and sea ice, but a bit more here would be good. For example: Is it easier to maintain? Is there performance benefits? Does the accurate simulation of surface albedo matter for climate given the small differences between the algorithms here? The text also needs some significant grammar checking. Here are some more specific suggestions:

1. It is very confusing when the delta-Eddington radiative transfer scheme of Breigleb and Light (2007) is interchangeably referred to as CICE, Icepack, delta-Eddington, adding-doubling delta-Eddington, etc. I suggest you just refer to it as dEdd everywhere and clearly explain that you are talking about the default implementation of Briegleb and Light (2007) and not some modified version?

2. The first sentence of the abstract should say something more quantitative than "large parts of the Earth".

3. The first two paragraphs should mention melt ponds and meltwater in the snow. These are critical for the seasonal cycle evolution of albedo.

4. Line 116: "This method has carried into the sea-ice ..." This is not proper usage.

5. The discussion of large solar Zenith angles is an interesting part. I think some suggestions of ways to improve how the models do this is needed.

6. Some mention of how the methods handle aerosols (black carbon and dust) would be good. For example, see Holland et al. 2012.

7. In the caption for Figure 1, I think you should spell out SWNB2. You refer to Figure 1 in the text before you define the acronym.

8. Figures 3-5 feel like they have a bunch of empty space (light red). You could almost cut off the panel axes below angles of 50 degrees.

9. The caption for Figure 4 is not grammatically correct and should be expanded.

10. The caption for Figure 5 is not correct. These panels are not the same as Figure

3. This caption should be expanded appropriately.

11. The caption for Figure 6 is not grammatically correct and should be expanded.

12. In general, I would prefer "two-stream" rather than "2-stream".

Holland et al. 2012: https://journals.ametsoc.org/doi/10.1175/JCLI-D-11-00078.1

---

## Author Comment (AC1) · 15 May 2019

Reply to the RCs: Inter-comparison and improvement of two-stream shortwave radiative transfer models for a unified treatment of cryospheric surfaces in ESMs

April 2019

We thank the two Reviewers for their insightful comments that have led to material improvements in the manuscript. To avoid confusion, we have replaced CICE with dEdd-AD to refer the radiative transfer core of sea-ice models; we have included one extra table (Table 1) to summarize the acronyms used in this work. Please find our replies to the questions and suggestions raised by the Reviewers below, and the revised manuscript with tracking changes in the supplement.

Anonymous Referee #1

This paper aims to unify the treatment of the optical properties of snow and ice in ESMs, because, historically, there have often been different albedo schemes in one and the same ESM, applied to seasonal snow, snow on sea ice, and snow on land ice masses like ice sheets. In particular, the authors focus on SNICAR (used for land ice and land masses), and Icepack/CICE (for sea ice). All radiative transfer schemes in ESMs are two-stream approximations. They assess the accuracy of the 2-stream approximations of different flavors against DISORT in a 16-stream configuration and consider this as the benchmark. I have few comments on the research itself. This is solid and has been carefully designed, and conducted. I do have critical remarks on the presentation of the results. They are surprisingly unclear in a few key parts, to the extent that I am unsure what the authors have actually done and what they haven't done.

The problem lies in the fact that there is (1) a correction for high SZA > 75 degrees, and (2) mention of a hybrid model SNICAR-AD. To me it is unclear whether these are the same thing or not. The high-SZA correction seems to be carried out on CICE rather than SNICAR. The authors suggest that a correction (equations 13 a and b) can be conducted for any 2-stream approximation, so also for SNICAR. So what is the situation after this paper? Do the authors now have 1 model for all snow and ice surfaces? Or did they present a correction for CICE only? Or also for SNICAR? And is this then SNICAR-AD? And what are the recommendation in section 8 about? Is this for future work? Or are these points that have been taken into account while a unified model framework was developed?

All in all, there are two possibilities: (1) either the authors have forgotten to mention that

the correction for SZA > 75 degrees is actually called SNICAR-AD. In that case, Figure 12 needs different figure axis labels and the text needs to clarify that this correction is called SNICAR-AD in a few places. In this case, I would suggest to move section 8 forward between the current sections 5 and 6, so as to present first the requirements for a unified model, and then the actual unified model. Or, (2) the correction for SZA > 75 degrees is not the same as SNICAR-AD, but rather an intermediate step in the unification of Icepack and SNICAR. In that case, the paper is incomplete. Results from the to-be-developed SNICAR-AD need to be incorporated here. If this would be paper 1 presenting the SZA-correction, and in a future paper 2 we will see SNICAR-AD fully developed, then I recommend that this paper is postponed to when SNICAR-AD is finished. I recommend to the editor to inquire with the authors which of these possible situations is the case, and then reach a decision.

Reply: We thank the Reviewer for their questions and suggestions. We have revised Sections 6 and 8 to address these concerns and to clarify our work.

First of all, SNICAR-AD is the unified treatment that we developed by merging SNICAR with dEdd-AD (note dEdd-AD was referred to as CICE in version 1, and is now called dEdd-AD in version 2 following the Reviewer's suggestion), and incorporating our correction for SZA > 75 degrees. We have finished the development of this scheme, and we are testing its performance using E3SM. To make this point clear, we added the following text in Section 8:

Lines 870-882: "We have merged these findings into a hybrid model SNICAR-AD, which is primarily composed of the radiative transfer scheme of dEdd-AD, 5-band snow/aerosol SSPs of SNICAR, and the parameterization to correct for snow albedo biases when solar zenith angle exceeds $75°$. This hybrid model can be applied to snow on land, land ice, and sea ice to produce consistent shortwave radiative properties for snow-covered surfaces across the Earth system. With the evolving and further understanding of snow and aerosol physics and chemistry, the adoption of this hybrid model will obviate the effort to modify and maintain separate optical variable input files used for different model components.

SNICAR-AD is now implemented in both the sea-ice (MPAS-seaice) and land (ELM) components of E3SM. More simulations and analyses are underway to examine its impact on E3SM model performance and simulated climate. The results are however beyond the scope of this work and will be thoroughly discussed in a future paper."

Second, the high-SZA correction applies in principle to any two-stream algorithms, including SNICAR and dEdd-AD (i.e. CICE in version 1). In Section 6, we introduced the high-SZA correction using dEdd-AD (i.e. CICE in version 1) assuming dEdd-AD will be the radiative transfer core used for all snow-covered surface per discussed in Section 8. The Reviewer's concern is however crucial since the adjustment factor R75+ is essentially a ratio of the exact reflectance to the two-stream reflectance, which is algorithm-specific,

$R_{(75+)} = \alpha_{(16\text{-}DISORT)}/\alpha_{(dEdd\text{-}AD)}$

We have included more discussion to make this point clear:

Lines 590-595: "For solar zenith angles $> 75°$, two-stream models underestimate snow albedo and overestimate solar absorption within snowpack, mostly in the top 2-cm of snow, and the differences among three two-stream models are small. In Section 5, we have shown that dEdd-AD produces the most accurate snow albedo in general, with anticipated wide application of dEdd-AD, we develop the following parameterization to adjust its low biases in computed near-IR direct albedo."

Lines 631-661: "When the solar zenith angle exceeds $75°$, our model adjusts the computed direct near-IR albedo $\alpha dEdd$-AD by the ratio R75+ following equations 12-14a and reduces direct near-IR absorption following equation 14b. If snow is divided into multiple layers, we assume all decreased near-IR absorption (2nd term on the right hand side, equation 14b) is confined within the top layer. This assumption is fairly accurate for the near-IR band, since most absorption occurs at the surface of snowpack (Figures 10 and 11). As discussed previously, this parameterization is developed based on albedo computed using dEdd-AD. For models that do not use dEdd-AD but SNICAR and 2SD, the same adjustment still applies given the small differences of near-IR direct albedo computed using two-stream models (Figure 11). For models that adopt other radiative transfer algorithms it is best for the developers to examine their model against a benchmark model such as 16-stream DISORT or two-stream models discussed in this work before applying this correction."

Thirdly, Section 8 summarizes the findings of this work, which are essentially the principles we follow to generate the merged model SNICAR-AD. We hope that the discussion and recommendations in this section are also useful to readers who are interested in improving their own snow radiative transfer schemes besides SNICAR or dEdd-AD. Some of the discussion includes future directions and features that are not yet included in SNICAR-AD, such as increasing the number of bands to match RRTMG. The revised text in section 8 reflects the above discussion.

Referee #2 David Bailey (dbailey@ucar.edu)

I have read the manuscript: "Inter-comparison and improvement of 2-stream shortwave radiative transfer models for a unified treatment of cryospheric surfaces in ESMs" by Dang et al. Overall, the article is interesting and may be of interest to the readers of the Cryosphere. I believe it needs some substantial revision before it would be acceptable however. There are a lot of model acronyms thrown around here and the text should be expanded to help clarify in some sections. Also, I feel it is lacking a bit in motivation. I understand from the title, the idea is to unify the radiative transfer schemes for snow on land, ice sheets, and sea ice, but a bit more here would be good. For example: Is it easier to maintain? Is there performance benefits? Does the accurate simulation of surface albedo matter for climate given the small differences between the algorithms here? The text also needs some significant grammar checking.

Reply: Thank you for these suggestions and questions.

To help clarify these models, we add a new table (Table 1 in the revised manuscript) to summarize the acronyms and their corresponding references used in this work.

Yes. It is easier to maintain an ESM with the unified scheme than with distinct schemes in the land and sea-ice components. For example, CICE-based sea-ice models utilize hardcoded snow single-scattering properties that are not easy to update, and that yield different reflectance and heating than SNICAR-based properties. With this unified treatment, an Earth system model only needs to maintain and update a single input optical data file shared by both land and sea-ice components. As our evaluations show, the adoption of dEdd-AD radiative transfer core, SNICAR single-scattering properties, and the high solar zenith angle parameterization improves the modeled physics.

The accurate simulation of surface albedo matters despite the apparently small differences between the algorithms. For example, compared to dEdd-AD, SNICAR and 2SD overestimate the diffuse albedo of melting snow by 0.015 (Figure 6). In Greenland, the daily-averaged downward diffuse solar flux from May to September is 200W/m 2, and the mean cloud cover fraction is 80% (Figure 6, Dang et al., 2017). In this case, SNICAR and 2SD overestimate the reflected solar flux by 0.015 * 200* 0.8 $\sim$ 2.4 W/m2, which is enough to melt 10 cm SWE over all of Greenland from May to September. dEdd-AD also remediates self-compensating spectral biases (where visible and Near-IR biases are of opposite signs) present in the other schemes. Those spectral biases do not affect the broadband fluxes like the diffuse biases, but they nevertheless degrade proper feedbacks between snow/ice reflectance and heating. To better evaluate these impacts globally, we are now performing coupled E3SM simulations. The results will be discussed in a following paper.

More discussion is included in Section 8 to clarify these points.

Lines 525-521: "These relatively small differences between algorithms may still yield large impact on snowpack. For example, compared to dEdd-AD, SNICAR and 2SD overestimate the diffuse albedo by $\sim$0.015 for melting snow (Figure 6). In Greenland, the daily averaged downward diffuse solar flux from May to September is 200 W/m2, and the averaged cloud cover fraction is 80% (Figure 6, Dang et al., 2017). In this case, SNICAR and 2SD overestimate the reflected solar flux by 2.4 W/m2 per day – the amount of energy otherwise enough to melt 10 cm of snow water equivalent from May to September. dEdd-AD also remediates self-compensating spectral biases (where visible and Near-IR biases are of opposite signs) present in the other schemes. Those spectral biases do not affect the broadband fluxes like the diffuse biases, but they nevertheless degrade proper feedbacks between snow/ice reflectance and heating."

Lines 870-882: "We have merged these findings into a hybrid model SNICAR-AD, which is primarily composed of the radiative transfer scheme of dEdd-AD, 5-band snow/aerosol SSPs of SNICAR, and the parameterization to correct for snow albedo biases when solar zenith angle exceeds 75°. This hybrid model can be applied to snow on land, land ice, and sea ice to produce consistent shortwave radiative properties for snow-covered surfaces across the Earth system. With the evolving and further under-standing of snow and aerosol physics and chemistry, the adoption of this hybrid model will obviate the effort to modify and maintain separate optical variable input files used for different model components.

SNICAR-AD is now implemented in both the sea-ice (MPAS-seaice) and land (ELM) components of E3SM. More simulations and analyses are underway to examine its impact on E3SM model performance and simulated climate. The results are however beyond the scope of this work and will be thoroughly discussed in a future paper"

Here are some more specific suggestions:

1. It is very confusing when the delta-Eddington radiative transfer scheme of Briegleb and Light (2007) is interchangeably referred to as CICE, Icepack, delta-Eddington, adding-doubling delta-Eddington, etc. I suggest you just refer to it as dEdd everywhere and clearly explain that you are talking about the default implementation of Briegleb and Light (2007) and not some modified version? Reply: Thank you for this suggestion.

We agree that using dEdd for sea-ice component is better than CICE since this is also the name of sea-ice radiative transfer scheme defined in Icepack/CICE/MPAS-seaice model namelist. The use of dEdd everywhere may, however, raise another confusion since SNICAR also adopts two-stream delta-Eddington approximation for snow visible optical properties, but with a different technique per discussed in Section 2. To distinguish the scheme of Briegleb and Light (2007) from what is implemented in SNICAR, we suggest referring to it as dEdd-AD, where AD is short for adding-doubling and corresponds to the AD in the name of the unified model SNICAR-AD. We have revised the related text, figures, and tables to reflect this change. We apply the default implementation of Briegleb and Light (2007) to snow in this work, which is stated in Section 2.2.

2. The first sentence of the abstract should say something more quantitative than "large parts of the Earth". Reply: Thank you. We have revised it as "mid and high latitudes of the Earth".

3. The first two paragraphs should mention melt ponds and meltwater in the snow. These are critical for the seasonal cycle evolution of albedo. Reply: Agreed. We now discuss snow melt in the first paragraph. Lines 46-50: "The accumulation, evolution, and depletion of snow cover modify the seasonal cycle of surface albedo globally. In particular, snow over sea ice absorbs more solar energy and begins to melt in the spring, which forms melt ponds that bring the sea-ice albedo to as low as 0.15 to further accelerate ice melt (Light et al., 2008, 2015)."

4. Line 116: "This method has carried into the sea-ice..." This is not proper usage. Reply: We have modified this sentence: Lines 127-130: "dEdd-AD has been adopted by the sea-ice physics library Icepack (https://github.com/CICE-Consortium/Icepack/wiki), which is used by the Los Alamos Sea Ice Model CICE (Hunke et al., 2010) and Model for Prediction Across Scales Sea Ice MPAS-seaice (Turner et al., 2018)."

5. The discussion of large solar Zenith angles is an interesting part. I think some suggestions of ways to improve how the models do this is needed. Reply: Thank you for the suggestion. We have modified section 6 to include more details on how to implement the adjustment when the solar zenith angles exceed 75 degrees: Lines 631-661: "When the solar zenith angle exceeds 75°, our model adjusts the computed direct near-IR albedo $\alpha$dEdd-AD by the ratio R75+ following equations 12-14a and reduces direct near-IR absorption following equation 14b. If snow is divided into multiple layers, we assume all decreased near-IR absorption (2nd term on the right hand side, equation 14b) is confined within the top layer. This assumption is fairly accurate for the near-IR band, since most absorption occurs at the surface of snowpack (Figures 10 and 11). As discussed previously, this parameterization is developed based on albedo computed using dEdd-AD. For models that do not use dEdd-AD but SNICAR and 2SD, the same adjustment still applies given the small differences of near-IR direct albedo computed using two-stream models (Figure 11). For models that adopt other radiative transfer algorithms it is best for the developers to examine their model against a benchmark model such as 16-stream DISORT or two-stream models discussed in this work before applying this correction."

6.    Some mention of how the methods handle aerosols (black carbon and dust) would be good.    For example, see Holland et al.    2012: https://journals.ametsoc.org/doi/10.1175/JCLI-D-11-00078.1 Reply: Thank you for this suggestion and reference. Aerosol-in-snow/ice is definitely one of the topics we are interested in. Currently, we are performing fully coupled E3SM simulations to quantify the radiative effects of aerosols in the cryosphere with SNICAR-AD. The analysis of these modeled results will serve the same purpose as Holland et al., 2012. We anticipate some change in the radiative effects of aerosols, but the unified model SNICAR-AD does not change how the snow model treats aerosols. The discussion can be found towards the end of Section 8:

Lines 843-857: "Both dEdd-AD and SNICAR simulate the impact of light-absorbing particles (black carbon and dust) on snow and/or sea ice using self-consistent particle

SSPs that follow the SNICAR convention (e.g., Flanner et al., 2007; Holland et al. 2012). These particles are assumed to be either internally or externally mixed with snow crystals; the combined SSPs of mixtures (e.g. Appendix A of Dang et al., 2015) are then used as inputs to the radiative transfer calculation. The adoption of dEdd-AD radiative transfer algorithm in SNICAR, and the implementation of SNICAR snow SSPs in dEdd-AD enables a consistent simulation of the radiative effects of light-absorbing particles in the cryosphere across ESM components."

7. In the caption for Figure 1, I think you should spell out SWNB2. You refer to Figure 1 in the text before you define the acronym. Reply: Thank you for pointing out this error. We have modified the text in Section 4 such that the original Figure 1 is not referred until SWNB2 is properly defined in the text. Note that we have switched the order of original Figures 1 and 2 since the Figure 2 in previous manuscript was the first figure cited in the text.

8. Figures 3-5 feel like they have a bunch of empty space (light red). You could almost cut off the panel axes below angles of 50 degrees. Reply: We would agree with the Reviewer if Figures 7-8 were not included in this paper. Figures 7-8 show the errors in reflected shortwave flux given albedo errors shown in Figures 3-4, while the errors in reflected flux varies with solar zenith angles. We prefer to keep the axes of Figures 3-5 as is since it is more straightforward for cross-comparison.

9. The caption for Figure 4 is not grammatically correct and should be expanded. Reply: Thank you for pointing this out. We have revised and expanded the figure caption.

10. The caption for Figure 5 is not correct. These panels are not the same as Figure 3. This caption should be expanded appropriately. Reply: Thank you for pointing this out. We have revised and expanded the figure caption.

11. The caption for Figure 6 is not grammatically correct and should be expanded. Reply: Thank you for pointing this out. We have revised and expanded the figure caption.

12. In general, I would prefer "two-stream" rather than "2-stream". Reply: We agree. We have replaced 2-stream with two-stream in the manuscript. Thank you for this suggestion.

Please also note the supplement to this comment:
https://www.the-cryosphere-discuss.net/tc-2019-22/tc-2019-22-AC1-supplement.pdf

**Supplement:**

[revised manuscript text omitted]

Author

Author

---

## Author Response (AR2)

**Reply to the Reviewers: Inter-comparison and improvement of 2-stream shortwave radiative transfer models for unified treatment of cryospheric surfaces in ESMs**

Jun 2019

We thank the Editor and two Reviewers for their additional comments. We have addressed the comments raised by the Reviewers as follows:

**Anonymous Referee #1**

I have examined the manuscript by Dang et al. and my assessment is that the paper would be suitable for publication in The Cryosphere after minor corrections.

The authors have taken away most of the issues raised by the reviewers, and clarified and expanded the paper in a few critical sections.

Remaining minor issues:

line 12: Following reviewer 2, please change "large parts of the Earth" to "mid- and high latitudes."
Fixed.

line 74: ungrammatical: please change to: "... ice is much more absorptive, so that the broadband (or: near-infrared) snow albedo is lower than the visible albedo".
Fixed line 96: in the literature list, you use Kuipers Munneke et al, but here you use Munneke et al.. Please make consistent.
Fixed line 114: multi-year snowpack -> a multi-year snowpack
Fixed line 146: at representing -> for representing
Fixed line 193: matter of taste but I would say: "where mu is the cosine of the zenith angle"
Fixed line 226: their algorithms -> its algorithms
Fixed line 354: to snowpack -> to the snowpack
Fixed line 433: Please change the order of figure 1 and 2. In the text you present first figure 2 and then 1. Also, Figure 2 would be a more introductory figure, so better to make it figure 1.

We agree that Figure 2 would be a more introductory figure, so we edited the text to cite Figure 2 first, and switched the order of Figures 1 and 2.

line 524: suggest: "Although the differences between algorithms are small, they can have a notable impact on snowpack melt. For example, ..."

Thanks for the suggestion, we have adopted this in the paper.

line 531: self-compensating -> compensating

Fixed line 569: please avoid this construction with opposing arguments in brackets. It is very confusing. Better: "two-stream models underestimate the column solar absorptions for new snow, and they overestimate them for old snow. ..."

Fixed line 594: within snowpack -> within the snowpack

Fixed line 595: among three two-stream models -> among the three two-stream models.

Fixed line 603: Break up this sentence into two sentences.

Fixed line 624: I assume that this correction is applied only for theta > 75 degrees. Please add this in the formula (for theta>75 deg)

We have edited Equation 12 to indicate this condition, which also applies to equations 14 (a) and (b).

$$R_{75+} = \frac{\alpha_{16-DISORT}}{\alpha_{dEdd-AD}} = c_1(\mu_0)log_{10}(r) + c_0(\mu_0), \text{ for } \mu_0 < 0.26, \text{ i.e. } \theta_0 > 75° \qquad (12)$$

line 654: surface of snowpack -> surface of the snowpack

Fixed line 663: suggest "Although the errors of direct near-IR albedos...."

Fixed line 666: can be important at/for high latitudes.

Fixed line 667: remains large -> is large
Fixed line 672: broader band-averaged. Please use consistent terms. Replace by "narrowband-averaged" or "band-averaged".
Fixed line 755: Replace the term "discussion" by "proposal for" or "requirements for" or "towards" or "designing".
We have replaced "discussion" with "towards". Thank you for this suggestion.

line 787: suggest "Third, in a cryospheric radiative transfer model, one should prefer ..."
Fixed line 839: Adopting dEdd-AD radiative core -> Adopting the dEdd-AD radiative core
Fixed

**Referee #2**
**David Bailey (dbailey@ucar.edu)**

The authors have generally addressed my concerns from my first review. I still do not like the first sentence of the abstract: "Snow is an important climate regulator because it greatly increases the surface albedo of large parts of the Earth." I am thinking something more like "15% or 20% of the Earth". It still kind of lacks in motivation, however the manuscript is generally fine and would be of interest to the readers of the Cryosphere.

Thank you. We have revised the abstract to:

[revised manuscript text omitted]